# GENERATIVE AND EXPLAINABLE DATA AUGMENTATION FOR SINGLE-DOMAIN GENERALIZATION

## ABSTRACT

In this work, we propose Generative and Explainable Adversarial Data Augmentation (GEADA), a novel framework designed to tackle the single-domain generalization challenge in image classification. The framework consists of two competing components: an augmentor to synthesize diverse yet semantically consistent augmentations, and a projector to learn domain-invariant representations from the augmented samples. The augmentor leverages a generative network for style transformations and an attribution-based cropping module for explainable geometric augmentations. We further incorporate theoretically-grounded contrastive loss functions, inspired by the geometric properties of unit hyperspheres, to promote the diversity of generated augmentations and the robustness of learned representations. Extensive experiments on multiple standard domain generalization benchmarks demonstrate the effectiveness of our approach against domain shifts.

## 1 INTRODUCTION

Deep neural networks (DNNs) have garnered remarkable success across diverse fields (LeCun et al., 2015; Silver et al., 2016; Radford et al., 2019) attributed to their exceptional generalization capabilities. However, this generalization is often predicated on the assumption that the training and test data originate from the same domain. In practice, the assumption is frequently compromised due to inevitable domain shifts caused by various factors such as domain mismatch (Sinha et al., 2018), data corruption (Hendrycks & Dietterich, 2019), and adversarial attacks (Madry et al., 2018), among others. Such discrepancies can lead to severe degradation in the performance of DNNs, thereby significantly undermining their safety and reliability, particularly in high-stakes applications such as autonomous driving and medical diagnosis.

To mitigate the performance deterioration induced by domain discrepancies, *domain adaptation* methods (Ganin & Lempitsky, 2015; Xu et al., 2019) and *multi-domain generalization* (MDG) approaches (Muandet et al., 2013; Shankar et al., 2018) have been proposed to enhance model generalization capabilities by leveraging predetermined test domains and multiple available training domains, respectively. Nevertheless, in real-world applications, acquiring additional test or training data from extra domains is often impractical due to constraints on the data acquisition budget. This limitation has spurred the advancement of *single-domain generalization* (SDG) techniques (Volpi et al., 2018; Qiao et al., 2020; Wan et al., 2022) that aim to promote model generalization on unseen domains solely based on training data from a single source.

Expanding upon this, *contrastive learning* (CL) approaches (Oord et al., 2018; Khosla et al., 2020) have been increasingly incorporated in domain generalization algorithms to learn domain-invariant feature representations (Kim et al., 2021), given their exceptional performance in both supervised and self-supervised learning scenarios. Essentially, CL algorithms are designed to learn invariant representations by simultaneously pulling together positive sample pairs and pushing apart negative samples in the embedding space. Particularly, the selection of positive samples emerges as a critical determinant of the quality of learned representations (Tian et al., 2020b). In the extant literature, positive views are commonly generated by applying stochastic data augmentations to the same source samples. However, stochastically augmented views can be either overly diverse, compromising critical task-relevant information, or excessively similar, leading to redundantly noisy representations, hence substantially hindering the generalization of learned representations in downstream tasks (Tian et al., 2020b; Peng et al., 2022).

In this paper, we concentrate on the SDG problem within the realm of image classification. To tackle the challenge, we propose a novel adversarial training framework termed *Generative and Explainable Adversarial Data Augmentation* (GEADA). The framework is comprised of two competing components: an Augmentor developed to synthesize diverse yet semantically consistent views, and a Projector tailored to learn robust representations from the augmented samples. Motivated by the empirical efficacy of *Color Jitter* and *Random Resized Crop* techniques in crafting CL positive views (Chen et al., 2020), we integrate both style and geometric augmentation in our framework. The principal contributions of our work can be summarized as follows.

- We devise a generative network equipped with style modulation layers, specifically designed to proficiently manipulate the color distribution of source samples by altering the statistics of their feature maps. By leveraging random style codes, this network is capable of synthesizing an arbitrary number of images that exhibit diverse styles while maintaining semantic consistency.

- We develop an explainable cropping technique that leverages a model interpretation approach to generate geometrically diverse views without sacrificing task-relevant information. Specifically, we uniformly select crop centers from distinct patches that are sampled from a multinomial distribution parameterized by the corresponding attribution scores, thereby statistically encouraging the cropped views to include various portions of task-relevant regions

- We propose theoretically justified loss functions for both data augmentation and representation learning by investigating the unit hypersphere geometry. Particularly, we initiate the *adversarial contrastive loss* to forster the diversity and semantic consistency of the augmented views by encouraging a uniform distribution of these views around the source sample in the embedding space. We also introduce the *supervised centroid loss* to learn domain-invariant representations by aligning the augmented views with the corresponding uniformly distributed class centroid on the unit hypersphere. The two loss functions are jointly optimized to enhance model generalization.

## 2 RELATED WORK

**Domain Generalization**. Unlike domain adaptation methods that are designed to align the training domain with predetermined test domains (Ganin & Lempitsky, 2015; Murez et al., 2018; Xu et al., 2019), MDG approaches (Muandet et al., 2013; Li et al., 2018; Shankar et al., 2018; Carlucci et al., 2019) operate without the prior information on the target domains and instead exploit multiple available source domains to learn domain-invariant representations. On the other hand, SDG algorithms are designed to address a more challenging yet realistic scenario where only a single training source is accessible. For instance, ADA (Volpi et al., 2018) augments the training domain by generating virtual images through adversarial updates on the input images. MixStyle (Zhou et al., 2020) integrates instance-level style mixing into the feature normalization layers to enhance model robustness. L2D (Wang et al., 2021) incorporates a style module to enrich image diversity through mutual information optimization. MetaCNN (Wan et al., 2022), meanwhile, takes a novel architectural approach, decomposing the convolutional features of images into meta-features to address the problem.

Similar to L2D (Wang et al., 2021), our framework also employs a style modulation network to implement style augmentation. However, in contrast to L2D, our framework is complemented by a theoretically-grounded loss function tailored for style diversity and semantic integrity, and a geometric augmentation technique to further facilitate robust representation learning.

**Contrastive Learning**. CL algorithms have consistently proven their efficacy across a diverse range of image classification tasks (Chen et al., 2020; He et al., 2020; Khosla et al., 2020). The pivotal ingredient in the achievement of CL algorithms lies in the specialized contrastive loss functions (Oord et al., 2018; Khosla et al., 2020), which effectively discriminate between positive and negative samples in the embedding space. Optimization of these loss functions can be conceptually understood as maximizing the mutual information (MI) between positive samples (Tian et al., 2020a). Subsequent studies by Wang & Isola (2020) and Chen et al. (2021) delve deeper into the nuanced properties of contrastive losses, exploring them from the dual perspectives of alignment and uniformity. Moreover, the construction of positive samples serves as another crucial factor underpinning the success of CL algorithms. Tian et al. (2020b) empirically demonstrate that MI between optimally chosen views should strike a delicate balance, excluding excessive noise while retaining task-specific information. Despite its critical role, the topic of view crafting remains relatively unexplored in the academic landscape, signposting an open avenue for future research.

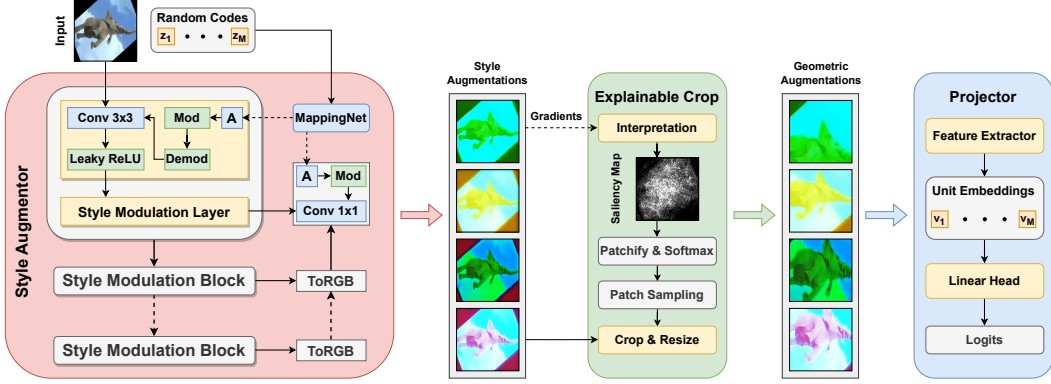

Figure 1: Overview of the GEADA framework. Style augmentations are initially generated by a style modulation network and then processed by the explainable cropping module to introduce geometric diversity. Afterwards, the projector extracts domain-invariant features from the augmented views.

Inspired by Wang & Isola (2020), we propose theoretically grounded contrastive loss functions, developed through an in-depth examination of the geometric properties of unit hyperspheres. Distinct from Wang & Isola (2020) and other CL-based domain generalization methods (Kim et al., 2021; Duboudin et al., 2021), our approach leverages the contrastive principles not only in the realm of representation learning but also in the process of view synthesis, thereby leading to diverse yet semantically consistent views that aligns well with the criteria demonstrated by Tian et al. (2020b).

**Model Interpretation**. To mitigate the risks associated with the black-box nature of DNNs, post-hoc interpretation techniques (Simonyan et al., 2013; Selvaraju et al., 2017; Lundberg & Lee, 2017) have been developed with the objective of elucidating the intricate internal mechanisms of the networks, all without requiring alterations to existing training algorithms or model architectures. Among these techniques, gradient-based attribution methods have garnered significant attention due to their simplicity and effectiveness. In the context of visual processing, these methods assign attribution scores to each pixel of the input image based on the gradient of the output with respect to the input. The attribution scores are then used to generate saliency maps that highlight the task-relevant regions of the input image. For example, *Naive Gradients* (Simonyan et al., 2013) directly calculates the attribution scores with the raw input gradients. Although straightforward, the corresponding saliency maps tend to be visually noisy with pixel attributions fluctuating sharply at small scales. To mitigate this, *SmoothGrad* (Smilkov et al., 2017) averages gradients over multiple perturbed samples in a neighborhood of input images. *Integrated Gradients* (Sundararajan et al., 2017) overcomes the gradient saturation problem by integrating the gradients along a linear path between the input image and a chosen reference, thereby providing a more comprehensive picture of feature importance.

In this paper, we leverage the gradient-based attribution method to develop an explainable cropping technique for geometrically diverse views. In contrast to *ContrastiveCrop* (Peng et al., 2022) that relies on upsampled feature maps from the final convolutional layer for semantic-aware cropping, our approach employs the averaged gradients across style-augmented views to calculate more precise saliency maps. Notably, our cropping technique can be seamlessly adapted to various architectures other than convolutional neural networks, such as Vision Transformers (Dosovitskiy et al., 2021).

## 3 METHODOLOGY

In the context of single domain generalization for a $K$-class classification task, let $\mathcal{S} = \{(\mathbf{x}_i, \mathbf{y}_i)\}_{i=1}^{N}$ denote the source domain with training samples $\mathbf{x}_i \in \mathcal{X} \subset \mathbb{R}^D$ and labels $\mathbf{y}_i \in \mathcal{Y} \subset [K] \coloneqq \{1, \cdots, K\}$. The primary objective of SDG is to train a classifier that generalizes well on unseen domains solely relying on $\mathcal{S}$. To achieve this, we propose a novel framework named *Generative and Explainable Adversarial Data Augmentation* (GEADA), which incorporates two core components:

- **Augmentor** comprises a generative model $G(\cdot)$ responsible for style augmentation, and a cropping module $\mathrm{Xcrop}(\cdot)$ for geometric augmentation. Specifically, the generative model $G$ incorporates

an MLP network $m : \mathcal{Z} \mapsto \mathcal{W}$ to map random vectors to intermediate style codes, and a style modulation network $g : \mathcal{X} \times \mathcal{W} \mapsto \mathcal{X}$ to transform the original images into style-augmented views. The cropping module $\mathrm{Xcrop} : \mathcal{X} \mapsto \mathcal{X}$ then applies cropping and resizing operations to these style-augmented views based on patchified saliency maps to introduce geometric diversity.

- **Projector** consists of a feature extractor $f : \mathcal{X} \mapsto \mathbb{S}^{d-1}$ that maps the input images onto a $d$-dimensional unit hypersphere, and a linear classification head $h : \mathbb{S}^{d-1} \mapsto \mathbb{R}^K$ that outputs the logits for the $K$-class classification.

As depicted in figure 1, the GEADA framework attains domain-invariant representations through a competitive interplay between the Augmentor and the Projector. The remainder of this section will delve into the key mechanisms and associated loss functions within the framework.

## 3.1 GENERATIVE STYLE AUGMENTATION

### 3.1.1 GENERATIVE NETWORK WITH STYLE MODULATION

Modification of spatial feature statistics has been validated as a remarkably effective approach for style transfer (Huang & Belongie, 2017), and such techniques have been further employed in generative models like the StyleGAN series (Karras et al., 2019; 2020; 2021) to generate high-quality images. Inspired by these advancements, we incorporate the style modulation layer from Style-GAN2 (Karras et al., 2020) into our style modulation network $g$ to implement style augmentation.

As illustrated in figure 1, the style modulation layer incorporates feature statistics modification and normalization into the conventional convolutional layer. Let $\{w_{ijk}\}$ represent the original convolutional weights, where $i$, $j$, and $k$ enumerate the input feature maps, output feature maps, and the spatial footprint of the convolution, respectively. The style modulation and demodulation operations can then be implemented by scaling the weights as

$$w'_{ijk} = \mathrm{s}_i \cdot w_{ijk}, \quad w''_{ijk} = \Big[ \sum_{i,k} (w'_{ijk})^2 + \epsilon \Big]^{-\frac{1}{2}} \cdot w'_{ijk},$$

where the $i$-th input channel is scaled by the style code $\mathrm{s}_i$ for modulation and the $j$-th output channel is then normalized for demodulation with a small constant $\epsilon$ to avoid numerical instability. The style code vector $\mathbf{s} = (\mathrm{s}_1, \cdots, \mathrm{s}_I) = A(\mathbf{w})$ is obtained via a learnable affine transformation $A$, where $\mathbf{w} = m(\mathbf{z})$ is an intermediate style code mapped from the standard Gaussian vector $\mathbf{z} \sim \mathcal{N}(\mathbf{0}, \mathbf{I})$.

Subsequently, we assemble the style modulation block by stacking two style modulation layers together. Every block is outfitted with a ToRGB layer, designed to generate triple-channel images across multiple resolutions. The intermediate images at lower resolutions are progressively upsampled and fused with their higher-resolution counterparts via skip connections to yield the final output. With such design, given any source image $\mathbf{x}_i$ and random vectors $\{\mathbf{z}_{ij}\}_{j=1}^{M}$, the generative model $G$ can synthesize $M$ augmented views $\mathbf{x}_{ij} = G(\mathbf{x}_i, \mathbf{z}_{ij}) = g(\mathbf{x}_i, m(\mathbf{z}_{ij}))$ with distinct styles.

### 3.1.2 ADVERSARIAL CONTRASTIVE LOSS

For the training samples $\{\mathbf{x}_i\}_{i=1}^{N}$, let $\{\mathbf{x}_{ij} = G(\mathbf{x}_i, \mathbf{z}_{ij})\}_{j=1}^{M}$ denote $M$ augmented views of sample $\mathbf{x}_i$ synthesized by the generative network $G$, and $\mathbf{v}_i = f(\mathbf{x}_i)$ and $\mathbf{v}_{ij} = f(\mathbf{x}_{ij})$ represent the corresponding embeddings on $\mathbb{S}^{d-1}$ projected by the feature extractor $f$. To simultaneously guarantee the diversity and semantic consistency of the augmented views, we propose the *Adversarial Contrastive Loss* (AdvCon Loss) for the generative network $G$ as

$$\mathcal{L}_{\mathrm{adv}} = \frac{1}{M} \sum_{i=1}^{N} \sum_{j=1}^{M} \big( \mathbf{v}_i^\top \mathbf{v}_{ij} - \gamma \big)^2 + \lambda_{\mathrm{adv}} \cdot \sum_{i=1}^{N} \log \sum_{j_1 < j_2}^{M} \exp \big( \mathbf{u}_{ij_1}^\top \mathbf{u}_{ij_2} / \tau_{\mathrm{adv}} \big), \tag{1}$$

where $\mathbf{u}_{ij} \coloneqq \mathbf{v}_{ij}^\perp / \|\mathbf{v}_{ij}^\perp\|$ with $\mathbf{v}_{ij}^\perp = \mathbf{v}_{ij} - (\mathbf{v}_i^\top \mathbf{v}_{ij}) \cdot \mathbf{v}_i$ denoting the orthogonal component of $\mathbf{v}_{ij}$ with respect to $\mathbf{v}_i$, and $\gamma$, $\lambda_{\mathrm{adv}}$ and $\tau_{\mathrm{adv}}$ are positive tuning parameters.

From the perspective of alignment and uniformity, the first term on the right-hand side of Equation 1 aligns the semantics of the augmented views and the source sample by restricting the $L^2$ distance

between their embeddings to $r = \sqrt{2 - 2\gamma}$. On the other hand, the second term fosters the diversity of views by minimizing the pairwise Gaussian potential (Wang & Isola, 2020) for unit vectors as

$$G_t(\boldsymbol{u}, \boldsymbol{v}) = e^{-||\boldsymbol{u}-\boldsymbol{v}||^2/2t} = e^{-1/t} \cdot \exp(\boldsymbol{u}^\top \boldsymbol{v}/t) \quad \text{for} \quad \boldsymbol{u}, \boldsymbol{v} \in \mathbb{S}^{d-1} \text{ and } t > 0,$$

which further fosters a uniform distribution of $\mathbf{u}_{ij}$ as explained in the following proposition.

**Proposition 1.** *For any $\boldsymbol{v}_0 \in \mathbb{S}^{d-1}$ and $\mathbb{S}_r(\boldsymbol{v}_0) \coloneqq \{\boldsymbol{v} \in \mathbb{R}^d : ||\boldsymbol{v} - \boldsymbol{v}_0|| = r\}$ with $0 < r < 2$, let $\mathbb{S}_r^*(\boldsymbol{v}_0) \coloneqq \{\boldsymbol{u} \in \mathbb{S}_r(\boldsymbol{v}_0) \cap \mathbb{S}^{d-1}\}$ denote the non-empty intersection between the two hyperspheres. Then the normalized surface area measure $\sigma$ inducing uniform distribution on $\mathbb{S}_r^*(\boldsymbol{v}_0)$ is the unique solution to*

$$\min_{\mu \in \mathcal{M}(\mathbb{S}_r^*(\boldsymbol{v}_0))} \int \int G_t\left(\mathrm{noc}(\boldsymbol{u}_1), \mathrm{noc}(\boldsymbol{u}_2)\right) \mathrm{d}\mu(\boldsymbol{u}_1) \, \mathrm{d}\mu(\boldsymbol{u}_2),$$

*where $\mathcal{M}(\mathbb{S}_r^*(\boldsymbol{v}_0))$ denotes the set of Borel probability measures on $\mathbb{S}_r^*(\boldsymbol{v}_0)$, $G_t$ denotes the pairwise Gaussian potential, and $\mathrm{noc}(\boldsymbol{u}) = (\boldsymbol{u} - (\boldsymbol{v}_0^\top \boldsymbol{u}) \cdot \boldsymbol{v}_0)/||\boldsymbol{u} - (\boldsymbol{v}_0^\top \boldsymbol{u}) \cdot \boldsymbol{v}_0||$ represents the normalized orthogonal component of $\boldsymbol{u}$ with respect to $\boldsymbol{v}_0$. Moreover, the normalized counting measures associated with the sequence of $N$ point minimizers of $G_t$ converge weak\* to $\sigma$.*

**Remark 1.** *Notice that $G_t\left(\mathrm{noc}(\boldsymbol{u}_1), \mathrm{noc}(\boldsymbol{u}_2)\right) = c_r \cdot G_t(\boldsymbol{u}_1, \boldsymbol{u}_2)$ for some constant $c_r$ under the condition $||\boldsymbol{u}_1 - \boldsymbol{v}_0|| = ||\boldsymbol{u}_2 - \boldsymbol{v}_0|| = r$, which does not generally hold without the constraint. As the distances between the augmented and source embeddings are not universally equal even under the alignment restriction, we exploit the normalized orthogonal components $\mathbf{u}_{ij}$ in Equation 1 to pursue the uniformity on the hyperplane perpendicular to $\boldsymbol{v}_i$ to eliminate the effect from the alignment term.*

The conclusions in Proposition 1 can be achieved through establishing a Borel isomorphism Srivastava (2008) between $\mathbb{S}_r^*(\boldsymbol{v}_0)$ and the unit hypersphere. The full proof is elaborated upon in Appendix A. Proposition 1 indicates that the diverse and uniformly-distributed embeddings can be achieved by minimizing the Gaussian potential under the constraint of $L^2$ distance.

Consequently, by harmonizing the alignment and uniformity regularization, the proposed AdvCon loss encourages the generative model to synthesize diverse yet semantically coherent augmentations with embeddings uniformly distributed on the surface of a small neighborhood of the source embeddings. Refer to figure 2 for an intuitive geometric interpretation of the AdvCon loss on $\mathbb{S}^2$, where the augmented views $\{\mathbf{v}_{1j}\}$ are encouraged to be uniformly distributed on a circle in the vicinity of the source embedding $\mathbf{v}_1$.

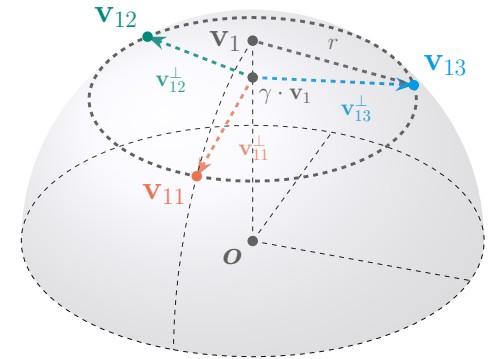

Figure 2: An illustration of the AdvCon loss on $\mathbb{S}^2$. The alignment term restricts $||\mathbf{v}_1 - \mathbf{v}_{1j}||$ to $r = \sqrt{2 - 2\gamma}$, while the uniformity term encourages the orthogonal component $\mathbf{v}_{1j}^\perp$ to be uniformly distributed on the circle centered at $\gamma \cdot \mathbf{v}_1$.

## 3.2 EXPLAINABLE GEOMETRIC AUGMENTATION

As highlighted by Chen et al. (2020), augmentations confined to the color space are not sufficient for complex CL tasks, and the integration of geometric augmentations can markedly boost the quality of learned representations. Among the spectrum of geometric augmentations, *RandomCrop*[1] has empirically validated its effectiveness across a multitude of applications, where the cropping center is uniformly sampled on the source image and the cropping dimensions are determined by the random area scale and aspect ratio. Nevertheless, as discussed by Peng et al. (2022), the uniformly cropped regions can either include redundant noise due to substantial overlapping or exclude critical task-relevant information due to excessive divergence.

To impose geometric diversity without sacrificing crucial information, we propose an explainable cropping module named *XCrop* that leverages the gradient-based attributions to guide the selection of cropped regions, ensuring they encompass diverse yet task-relevant regions. For views $\{\mathbf{x}_{ij} =$

---

[1]We omit the resize operation here as it is mandatory to align input dimensions.

$G(\mathbf{x}_i, \mathbf{z}_{ij})\}_{j=1}^M$, let $h_{\mathbf{y}_i}(\mathbf{v}_{ij})$ denote the logit corresponding to the true label $\mathbf{y}_i$, which is linearly transformed from the embedding $\mathbf{v}_{ij} = f(\mathbf{x}_{ij})$. The attribution score for sample $\mathbf{x}_i$ is computed as

$$\mathrm{Attr}(\mathbf{x}_i) = \frac{1}{M} \sum_{j=1}^M \nabla_{\mathbf{x}_{ij}} h_{\mathbf{y}_i}(f(\mathbf{x}_{ij})).$$

Notice that *SmoothGrad* (Smilkov et al., 2017) averages the gradients over the perturbed samples $\mathbf{x}_i + \boldsymbol{\epsilon}_{ij}$ with $\boldsymbol{\epsilon}_{ij} \sim \mathcal{N}(\mathbf{0}, \sigma^2 \cdot \boldsymbol{I})$. However, an improperly selected $\sigma$ can result in the score function $h_{\mathbf{y}_i}$ remaining static or exhibiting high fluctuations across the perturbed samples. In contrast, our method averages the gradients across the style-augmented views, which are generated to have representations uniformly distributed around the source embedding. Therefore, the logits $h(\mathbf{v}_{ij})$ are expected to exhibit moderate variance, thereby leading to substantially robust interpretation results.

Subsequently, we proceed to compute the saliency map based on the attribution scores $\mathrm{Attr}(\mathbf{x}_i)$. We first average the absolute values across the channel dimension, and then zero out the values that fall below the $\lambda_{\mathrm{crop}}$-th quantile of each map. The saliency map is further evenly divided into $P$ patches with $s_j(\mathbf{x}_i)$ denoting the sum of saliency scores within the $j$-th patch. For $M$ style-augmented views, we sample $M$ distinct patch locations from a multinomial distribution without replacement, and then uniformly draw the cropping center for each view from the correspondingly sampled patch. The multinomial distribution is parameterized by the saliency-based probability vector as

$$\boldsymbol{q} = (q_1, \cdots, q_P) = \mathrm{Softmax}\left( \frac{s_1(\mathbf{x}_i)}{\tau_{crop}}, \cdots, \frac{s_P(\mathbf{x}_i)}{\tau_{crop}} \right),$$

where $\tau_{crop}$ is the tuning parameter. Finally, the explainable cropping module can be formulated as

$$\tilde{\mathbf{x}}_{ij} = \mathrm{XCrop}(\mathbf{x}_{ij}, p_j, s_j, r_j),$$

where $p_j$ specifies the sampled patch location, and $s_j$ and $r_j$ represent the random area scale and aspect ratio, respectively. The explainable module enhances the geometric diversity of style-augmented views by ensuring that each view contains different proportions of task-relevant regions. For computational efficiency, it is practical to store the probability vector $\mathbf{q}$ for each sample and update it at periodic intervals, thereby reducing the overhead associated with gradient calculations.

## 3.3 SUPERVISED REPRESENTATION LEARNING

The remaining objective is to learn domain-agnostic representations for classification from the augmented views $\{\tilde{\mathbf{x}}_{ij} = \mathrm{XCrop}(G(\mathbf{x}_i, \mathbf{z}_{ij}))\}_{i,j=1}^{N,M}$. Let the linear head $h$ be parameterized by the normalized weight matrix $\mathbf{W} = (\mathbf{w}_1, \ldots, \mathbf{w}_K) \in \mathbb{R}^{d \times K}$, where $\mathbf{w}_k \in \mathbb{R}^d$ denotes the $k$-th column satisfying $||\mathbf{w}_k||_2 = 1$. Under this formulation, $\mathbf{w}_k$ can be conceptualized as the centroid of class $k$ on the unit hypersphere $\mathbb{S}^{d-1}$. In order to encourage representations to align with their corresponding class centroids, we introduce the *Supervised Centroid Loss* (SupCent Loss) of the form

$$\mathcal{L}_{\mathrm{sup}} = -\frac{1}{M} \sum_{i=1}^N \frac{1}{K_i} \sum_{j=1}^M \log \frac{\exp(\tilde{\mathbf{v}}_{ij}^\top \mathbf{w}_{\mathbf{y}_i}/\tau_{\mathrm{sup}})}{\sum_{k=1}^K \exp(\tilde{\mathbf{v}}_{ij}^\top \mathbf{w}_k/\tau_{\mathrm{sup}})} + \lambda_{\mathrm{sup}} \cdot \log \sum_{k_1 < k_2}^K \exp\left( \mathbf{w}_{k_1}^\top \mathbf{w}_{k_2}/\tau_{\mathrm{sup}} \right), \quad (2)$$

where $\tilde{\mathbf{v}}_{ij} = f(\tilde{\mathbf{x}}_{ij})$ denotes the embedding of augmented view $\tilde{\mathbf{x}}_{ij}$, $K_i = \#\{j \in [N] : \mathbf{y}_j = \mathbf{y}_i\}$ denotes the number of samples belong to class $\mathbf{y}_i$, and $\tau_{\mathrm{sup}}$ and $\lambda_{\mathrm{sup}}$ are some tuning parameters.

Intuitively, the first term on the right-hand side of Equation 2 aims to pull augmented embeddings towards the centroids of their respective ground-truth classes, while simultaneously pushing them away from the centroids of other classes. Concurrently, the second term encourages the class centroids to be uniformly distributed on $\mathbb{S}^{d-1}$ through minimizing the pairwise Gaussian potential as proved by Wang & Isola (2020). These two terms are jointly optimized, with the objective of clustering embeddings from different classes onto disjoint hyperspherical caps, to facilitate the subsequent classification capitalizing on the inherent linear separability of unit hyperspheres (See Appendix B).

In contrast to the supervised contrastive loss (Khosla et al., 2020), the SupCent loss focuses on the interplay between individual embeddings and the class centroids, rather than the interactions among embeddings belonging to the same class. The strategy can significantly improve computational efficiency by minimizing the cost of data communication across multiple devices in distributed training. Moreover, the SupCent loss reduces to the cross-entropy loss under the condition $\tau_{\mathrm{sup}} = 1$ and $\lambda_{\mathrm{sup}} = 0$ when assuming a balanced class distribution. However, the second term in the SupCent loss, which promotes uniform class weights, substantially improve the linear separability of learned representations. The empirical efficacy of the SupCent loss is further demonstrated in Section 4.2.

## 4 EXPERIMENTS

### 4.1 DOMAIN GENERALIZATION PERFORMANCE

**Baselines**. We evaluate the GEADA framework on multiple domain generalization benchmarks, including Digits, CIFAR-10-C, CIFAR-100-C (Hendrycks & Dietterich, 2019), PACS (Li et al., 2017) and Office-Home (Venkateswara et al., 2017). We validate its efficacy through a relatively comprehensive comparison with established baselines, including ERM baseline (Vapnik, 1999), data augmentation baselines Mixup (Zhang et al., 2018), AutoAug (Cubuk et al., 2019) and AugMix (Hendrycks et al., 2019), and domain generalization baselines JiGen (Carlucci et al., 2019), ADA (Volpi et al., 2018), MixStyle (Zhou et al., 2020), RSC (Huang et al., 2020), RandConv (Xu et al., 2021), L2D (Wang et al., 2021), MetaCNN (Wan et al., 2022), SelfReg (Kim et al., 2021), SagNet (Nam et al., 2021), XDED (Lee et al., 2022) and ALT (Gokhale et al., 2023).

**Setups**. For fair comparison, we adopt the same training and evaluation protocols as most existing work, and only present the results reported in the original papers. Particularly, for Digits and CIFAR-based benchmarks, the explainable cropping module is not implemented due to small image sizes, while both style and geometric augmentations are applied to the PACS and Office-Home benchmarks. Moreover, for the GEADA framework, we build the feature extractor $f$ based on specific model architectures with the output dimension adjusted to $d = 128$ for projection, and the mapping network $m$ is chosen as a 6-layer MLP network with 512 hidden units. Then number of views $M$ is set to 4 across all benchmarks. Additional training details can be found in Appendix C.

#### 4.1.1 DIGITS BENCHMARK

**Datasets**. The Digits benchmark comprises digit images sourced from five distinct domains: MNIST (LeCun et al., 1989), MNIST-M (Ganin & Lempitsky, 2015), SYN (Ganin & Lempitsky, 2015), SVHN (Netzer et al., 2011), and USPS (Denker et al., 1988). In adherence to the established conventions (Volpi et al., 2018; Wang et al., 2021), we take the first $10,000$ training images of MNIST as source samples and evaluate model performance on the test sets of the remaining datasets. For the GEADA framework, we employ a feature extractor $f$ based on LeNet (LeCun et al., 1989), and the generative network $g$ consists of 2 style modulation blocks with a base channel of $8$.

**Results**. Domain generalization performance on the Digits benchmark is encapsulated in Table 1. Remarkably, GEADA surpasses all competing methods, achieving an average accuracy of $80.80\%$, which constitutes a remarkable $2.04\%$ improvement over the previous state-of-the-art. Importantly, GEADA exhibits marked improvements in classification accuracies specifically on MNIST-M, SYN, and SVHN datasets, which are characterized by moderate to high domain shifts relative to the MNIST dataset. Meanwhile, the framework also achieves remarkable performance on USPS, which has a relatively small domain gap to the training source.

Table 1: Domain generalization accuracy (%) on the Digits benchmark. Trained on the first $10,000$ training samples of MNIST and evaluated on rest testsets.

| Method | MNIST-M | SVHN | SYN | USPS | Avg. |
|---|---|---|---|---|---|
| ERM | 52.72 | 27.83 | 39.65 | 76.94 | 49.29 |
| JiGen | 57.80 | 33.80 | 43.79 | 77.15 | 53.14 |
| AugMix | 53.36 | 25.96 | 42.90 | 96.12 | 54.59 |
| ADA | 60.41 | 35.51 | 45.32 | 77.26 | 54.62 |
| RandConv | 87.76 | 57.52 | 62.88 | 83.36 | 72.88 |
| ALT | 75.98 | 55.01 | 69.93 | **96.17** | 74.27 |
| L2D | 87.30 | 62.86 | 63.72 | 83.97 | 74.46 |
| MetaCNN | **88.27** | 66.50 | 70.66 | 89.64 | 78.76 |
| GEADA | 86.53 | **67.74** | **73.12** | 95.81 | **80.80** |
| (std) | (1.17) | (0.93) | (0.86) | (0.55) | |

#### 4.1.2 CIFAR-10-C AND CIFAR-100-C BENCHMARKS

**Datasets**. Both CIFAR-10-C and CIFAR-100-C benchmarks (Hendrycks & Dietterich, 2019) are specifically designed to assess model robustness against common types of image corruptions. These benchmarks are derived from the original CIFAR-10 and CIFAR-100 datasets (Krizhevsky, 2009), which contain $50,000$ training images and $10,000$ test images of $32 \times 32$ pixel dimensions from 10 and 100 classes, respectively. The benchmarks extend the test sets by introducing 15 different types of corruptions, each with five levels of severity. Here we use the uncorrupted training images as the source domain, and report the average classification error on the corrupted test samples across all severity levels. Moreover, we build the feature extractor $f$ based on WideResNet-40-2 (Zagoruyko & Komodakis, 2016), and the network $g$ includes 2 style blocks with a base channel of $16$.

Table 2: Single-domain accuracies (%) on PACS (A: Art Painting, C: Cartoon, S:Sketch, P:Photo).

| Method | A→C | A→S | A→P | C→A | C→S | C→P | S→A | S→C | S→P | P→A | P→C | P→S | Avg. |
|---|---|---|---|---|---|---|---|---|---|---|---|---|---|
| ERM | 62.3 | 49.0 | 95.2 | 65.7 | 60.7 | 83.6 | 28.0 | 54.5 | 35.6 | 64.1 | 23.6 | 29.1 | 54.3 |
| JiGen | 57.0 | 50.0 | 96.1 | 65.3 | 65.9 | 85.5 | 26.6 | 41.1 | 42.8 | 62.4 | 27.2 | 35.5 | 54.6 |
| MixStyle | 65.5 | 49.8 | 96.7 | 69.9 | 64.5 | 85.3 | 27.1 | 50.9 | 32.6 | 67.7 | 38.9 | 39.1 | 57.4 |
| RSC | 62.5 | 53.1 | 96.2 | 68.9 | 70.3 | 85.8 | 37.9 | 56.3 | 47.4 | 66.3 | 26.4 | 32.0 | 58.6 |
| SelfReg | 65.2 | 55.9 | 96.6 | 72.0 | 70.0 | 87.5 | 37.1 | 54.0 | 46.0 | 67.7 | 28.9 | 33.7 | 59.5 |
| SagNet | 67.1 | 56.8 | 95.7 | 72.1 | 69.2 | 85.7 | 41.1 | 62.9 | 46.2 | 69.8 | 35.1 | 40.7 | 61.9 |
| XDED | 74.6 | 58.1 | **96.8** | 74.4 | 69.6 | **87.6** | 43.3 | **65.6** | 50.3 | **71.4** | 54.3 | 51.5 | 66.5 |
| GEADA | **76.2** | **64.5** | 94.9 | **78.3** | **76.9** | 86.1 | **56.2** | 63.1 | **52.4** | 68.1 | **56.4** | **61.7** | **69.6** |

**Results**. Generalization results on the corrupted image benchmarks are listed in Table 3. Notably, GEADA achieves the second best performance, surpassing all competing methods except for AugMix (Hendrycks et al., 2019), which is the state-of-the-art method dedicated to data corruption. The success of AugMix can be partially attributed to the incorporation of a variety of data augmentation techniques, such as rotation and shear, which are not included in our framework. It motivates us to explore the potential of GEADA by introducing extra augmentations.

Table 3: Domain generalization error (%) on CIFAR-10-C and CIFAR-100-C across five severity levels.

| Method | CIFAR-10-C | CIFAR-100-C |
|---|---|---|
| ERM | 26.9 | 53.3 |
| Mixup | 22.3 | 50.4 |
| AutoAug | 23.9 | 49.6 |
| AugMix | **11.2** | **35.9** |
| XDED | 18.5 | 46.6 |
| GEADA | 14.3 | 42.8 |

### 4.1.3 PACS AND OFFICE-HOME BENCHMARKS

**Datasets**. PACS (Li et al., 2017) and Office-Home (Venkateswara et al., 2017) benchmarks have been created to evaluate model generalization capability on various domains. Specifically, PACS benchmark consists of totally $9,991$ images in 7 categories from four distinct domains, including *Photo*, *Art Painting*, *Cartoon* and *Sketch*. Meanwhile, Office-Home benchmark contains $15,500$ images from four domains, including *Artistic*, *Clip Art*, *Product* and *Real-World*, with each domain containing 65 object categories found typically in office and home settings. For PACS, we choose each training domain as the source domain and evaluate model performance on the remaining three domains separately. For Office-Home, we employ the leave-one-domain-out protocol, where one domain is selected as the test domain and the rest are treated as the source domain. For both benchmarks, the feature extractor is built on the ResNet-18 (He et al., 2016) backbone.

**PACS results**. SDG results on the PACS benchmark are summarized in Table 2, where GEADA achieves the best overall performance with a remarkable leading of $3.1\%$ compared to the previous state-of-the-art XDED. Notably, the improvement mainly comes from the performance boost regarding the challenging *Sketch* domain, where GEADA obtains $12.9\%$, $10.2\%$ and $6.6\%$ improvement in S→A, P→S and C→S scenarios, respectively. In general, GEADA achieves a more balanced performance across all domains, indicating its effectiveness against a wider range of domain shifts.

**Office-Home results**. In Table 4, the leave-one-domain-out performance on the Office-Home benchmark is reported. GEADA achieves the best overall performance with a modest but noteworthy improvement of approximately $0.5\%$. While the performance in less challenging cases (P and R) is slightly compromised, GEADA exhibits significant gains in more difficult scenarios (A and C). The promising performance of GEADA on both the CIFAR-100-C and Office-Home benchmarks affirms its efficacy, even in settings with a large number of classes.

Table 4: Leave-one-domain-out accuracies (%) on Office-Home (A:Artistic C:Clipart P:Product R:Real).

| Method | A | C | P | R | Avg. |
|---|---|---|---|---|---|
| ERM | 58.9 | 49.4 | 74.3 | 76.2 | 64.7 |
| JiGen | 53.0 | 47.4 | 71.4 | 72.7 | 61.2 |
| MixStyle | 58.7 | 53.4 | 74.2 | 75.9 | 65.5 |
| RSC | 58.4 | 47.9 | 71.6 | 74.5 | 63.1 |
| SagNet | 60.2 | 45.3 | 70.4 | 73.3 | 62.3 |
| XDED | 60.8 | 57.1 | 75.3 | 76.5 | 67.4 |
| GEADA | **62.3** | **59.2** | 74.7 | 75.2 | **67.9** |

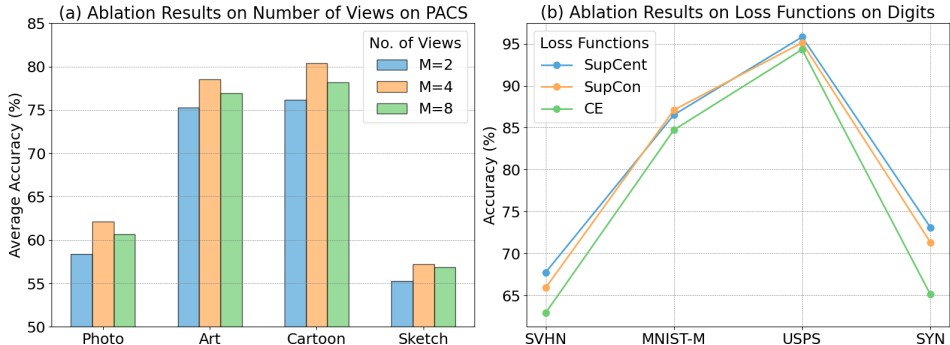

Figure 3: Single-domain generalization performance on (a) PACS benchmark with varying number of views and (b) Digits benchmark with different loss functions for representation learning.

## 4.2 ABLATION STUDIES

We proceed to conduct ablation studies to examine the effect of several critical factors in our framework, including the number of augmented views, choice of loss functions and the XCrop module.

**Number of augmented views**. We first investigate the influence of the number of augmented views $M$ on the PACS benchmark. We set the effective batch size, calculated as Batch Size $\times M$, to 512 during the experiment. Figure 3(a) shows that GEADA attains its peak performance when $M = 4$. We hypothesize that a small $M$ does not provide sufficient diversity for effective representation learning. Conversely, when $M$ is sufficiently large, further increasing $M$ may not yield additional diversity due to the uniformity of the views. Moreover, it may also dilute the class-specific information in each batch, given the constraints of the effective batch size.

**Loss function for representation learning**. As discussed in Section 3.3, the proposed SupCent loss shares conceptual similarities with the supervised contrastive (SupCon) loss and the cross-entropy (CE) loss. We then examine the effect of the loss function on the Digits benchmark. As depicted in Figure 3(b), the SupCent loss achieves significant improvement over the CE loss, possibly due to the uniformity of the class centroids. Moreover, the SupCent loss achieves comparable performance with the SupCon loss as both of them are designed to pull together embeddings from same class. However, the SupCent loss is more efficient by reducing the cost of data communication.

**Explainable cropping**. Following the setup in Section 4.1.3, we examine the merits of XCrop on the PACS and Office-Home benchmarks by comparing it with RandomCrop under the same settings. As shown in Table 5, GEADA significantly benefits from the attribution-based cropping on both benchmarks to preserve key information while making diverse augmentations.

Table 5: Average generalization accuracies (%) on PACS and Office-Home with RandCrop or XCrop.

| GEADA | PACS | Office-Home |
|---|---|---|
| RandCrop | 66.1 | 65.8 |
| XCrop | 69.6 | 67.9 |

## 5 DISCUSSION

In this paper, we present the GEADA framework to address the challenge of SDG in image classification tasks. Guided by the AdvCon and SupCent losses, our framework generates diverse yet semantically consistent style and geometric augmentations and learns domain-invariant features from them. In light of the framework, we focus on employing CL-based losses for data augmentation and representation learning, targeting two key augmentation types pivotal to image classification. However, as indicated in Section 4.1.2, introducing extra diverse augmentations could yield further improvements. Fortunately, the inherent flexibility of our framework allows for seamless incorporation of differentiable augmentation strategies. Future directions include extending the framework to embrace additional augmentation types, aiming to enhance generalization across diverse tasks in computer vision, audio processing, and natural language processing.

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

# A    PROOF OF PROPOSITION 1

The proof of Proposition 1 relies on the strict positive definiteness of the Gaussian kernel and the corresponding characterizations of the uniform distribution on $\mathbb{S}^{d-1}$ as demonstrated as follows.

**Lemma 1** (Borodachov et al. (2019); Wang & Isola (2020)). *For any $t > 0$, the Gaussian kernel*

$$G_t(\boldsymbol{u}, \boldsymbol{v}) = e^{-||\boldsymbol{u}-\boldsymbol{v}||^2/2t}$$

*is strictly positive definite on $\mathbb{S}^{d-1} \times \mathbb{S}^{d-1}$, i.e., if for any finite signed Borel measure $\mu$ on $\mathbb{S}^{d-1}$, the energy*

$$I_{G_t}[\mu] = \int_{\mathbb{S}^{d-1}} \int_{\mathbb{S}^{d-1}} G_t(\boldsymbol{u}, \boldsymbol{v}) \,\mathrm{d}\mu(\boldsymbol{u}) \,\mathrm{d}\mu(\boldsymbol{v})$$

*is well defined, we have $I_{G_t}[\mu] \geq 0$ and the equality holds only if $\mu(A) = 0$ for any $A \in \mathcal{B}(\mathbb{S}^{d-1})$.*

**Lemma 2** (Borodachov et al. (2019); Wang & Isola (2020)). *Consider kernel $K_f : \mathbb{S}^{d-1} \times \mathbb{S}^{d-1} \mapsto (-\infty, +\infty]$ of the form*

$$K_f(\boldsymbol{u}, \boldsymbol{v}) = f(||\boldsymbol{u} - \boldsymbol{v}||^2),$$

*which takes the Gaussian kernel as a special case. If $K_f$ is strictly positive definite on $\mathbb{S}^{d-1} \times \mathbb{S}^{d-1}$ and $I_{K_f}[\sigma_{d-1}]$ is finite, then the normalized surface area measure $\sigma_{d-1}$ is the unique minimizer of*

$$\min_{\mu \in \mathcal{M}(\mathbb{S}^{d-1})} I_{K_f}[\mu] = \min_{\mu \in \mathcal{M}(\mathbb{S}^{d-1})} \int_{\mathbb{S}^{d-1}} \int_{\mathbb{S}^{d-1}} K_f(\boldsymbol{u}, \boldsymbol{v}) \,\mathrm{d}\mu(\boldsymbol{u}) \,\mathrm{d}\mu(\boldsymbol{v}).$$

*Moreover, the normalized counting measures $\{\mu_N\}_{N=1}^{\infty}$ associated with the $N$-point $K_f$-energy minimizers $\{U_N^*\}_{N=1}^{\infty}$ given by*

$$U_N^* = \{\boldsymbol{u}_1^*, \cdots, \boldsymbol{u}_N^*\} = \operatorname*{arg\,min}_{\boldsymbol{u}_1, \cdots, \boldsymbol{u}_N \in \mathbb{S}^{d-1}} \sum_{1 \leq i < j \leq N} K_f(\boldsymbol{u}_i, \boldsymbol{u}_j)$$

*converge weak\* to $\sigma_{d-1}$, that is, for any continuous function $f_c : \mathbb{S}^{d-1} \mapsto \mathbb{R}$, we have*

$$\lim_{N \to \infty} \int_{\mathbb{S}^{d-1}} f_c(\boldsymbol{u}) \,\mathrm{d}\mu_N(\boldsymbol{u}) = \int_{\mathbb{S}^{d-1}} f_c(\boldsymbol{u}) \,\mathrm{d}\sigma_{d-1}(\boldsymbol{u}).$$

*Proof of Proposition 4.1.* Given any $\boldsymbol{v}_0 \in \mathbb{S}^{d-1}$ and $0 < r < 2$, to apply above results to the intersection $\mathbb{S}_r^*(\boldsymbol{v}_0) = \mathbb{S}_r(\boldsymbol{v}_0) \cap \mathbb{S}^{d-1}$, it is sufficient to establish the Borel isomorphism (Srivastava, 2008), a bimeasurable bijective mapping, between the intersection and a unit hypersphere.

In fact, with $\gamma = 1 - r^2/2$ and $r_\gamma = \sqrt{1 - \gamma^2}$, $\mathbb{S}_r^*(\boldsymbol{v}_0)$ is equivalent to a hypersphere centered at $\boldsymbol{v}_\gamma = \gamma \cdot \boldsymbol{v}_0$ with a radius of $r_\gamma$, restricted to the hyperplane $\mathbb{H}_\gamma(\boldsymbol{v}_0) = \{\boldsymbol{u} \in \mathbb{R}^d : \boldsymbol{u}^\top \boldsymbol{v}_0 = \gamma\}$. Specifically, for any $\boldsymbol{u} \in \mathbb{S}_r^*(\boldsymbol{v}_0)$, we have

$$||\boldsymbol{u} - \boldsymbol{v}_\gamma||^2 = ||\boldsymbol{u}||^2 + \gamma^2 ||\boldsymbol{v}_0||^2 - 2\gamma \cdot \boldsymbol{u}^\top \boldsymbol{v}_0 = 1 - \gamma^2.$$

On the other hand, for any $\boldsymbol{u} \in \mathbb{H}_\gamma(\boldsymbol{v}_0)$ satisfying $||\boldsymbol{u} - \boldsymbol{v}_\gamma||^2 = 1 - \gamma^2$, we also have $\boldsymbol{u} \in \mathbb{S}_r^*(\boldsymbol{v}_0)$.

Without loss of generality, we assume $\boldsymbol{v}_0 = \boldsymbol{e}_1 = (1, 0, \cdots, 0)^\top$, otherwise it can be achieved via the rotation of axes. Then, any $\boldsymbol{u} \in \mathbb{S}_r^*(\boldsymbol{v}_0)$ can be expressed as $\boldsymbol{u} = (\gamma, \boldsymbol{w}^\top)^\top$, where $\boldsymbol{w} \in \mathbb{R}^{d-1}$ satisfying $||\boldsymbol{w}||^2 = r_\gamma^2 = 1 - \gamma^2$. Consequently, there exists a bimeasurable bijective mapping $\phi : \mathbb{S}_r^*(\boldsymbol{v}_0) \mapsto \mathbb{S}^{d-2}$ given by

$$\phi(\boldsymbol{u}) = \phi((\gamma, \boldsymbol{w}^\top)^\top) = r_\gamma^{-1} \cdot \boldsymbol{w} = \tilde{\boldsymbol{w}} \quad \text{and} \quad \phi^{-1}(\tilde{\boldsymbol{w}}) = (\gamma, r_\gamma \cdot \tilde{\boldsymbol{w}}^\top)^\top = (\gamma, \boldsymbol{w}^\top)^\top = \boldsymbol{u},$$

which leads to the Borel isomorphism between $(\mathbb{S}_r^*(\boldsymbol{v}_0), \mathcal{B}(\mathbb{S}_r^*(\boldsymbol{v}_0)))$ and $(\mathbb{S}^{d-2}, \mathcal{B}(\mathbb{S}^{d-2}))$. Consequently, for any probability measure $\mu \in \mathcal{M}(\mathbb{S}^{d-2})$, let $\mu \circ \phi$ denote the pushforward measure of $\mu$ under $\phi$ on $(\mathbb{S}_r^*(\boldsymbol{v}_0), \mathcal{B}(\mathbb{S}_r^*(\boldsymbol{v}_0)))$, that is, for any $A \in \mathcal{B}(\mathbb{S}_r^*(\boldsymbol{v}_0))$, we have $\mu \circ \phi(A) = \mu(\phi(A))$. Analogously, for any $\nu \in \mathcal{M}(\mathbb{S}_r^*(\boldsymbol{v}_0))$, we can also define the pushforward measure $\nu \circ \phi^{-1}$ on $(\mathbb{S}^{d-2}, \mathcal{B}(\mathbb{S}^{d-2}))$. In particular, for the normalized surface area measures $\sigma$ and $\sigma_{d-2}$ on $\mathbb{S}_r^*(\boldsymbol{v}_0)$ and $\mathbb{S}^{d-2}$, respectively, we have $\sigma = \sigma_{d-2} \circ \phi$ and $\sigma_{d-2} = \sigma \circ \phi^{-1}$.

Additionally, let $G_t$ and $\tilde{G}_t$ denote the Gaussian kernels on $\mathbb{R}^d$ and $\mathbb{R}^{d-1}$, respectively. Then for any $\boldsymbol{u}_1, \boldsymbol{u}_2 \in \mathbb{S}_r^*(\boldsymbol{v}_0)$ with $\tilde{\boldsymbol{w}}_1 = \phi(\boldsymbol{u}_1), \tilde{\boldsymbol{w}}_2 = \phi(\boldsymbol{u}_2) \in \mathbb{S}^{d-2}$, we have

$$G_t(\boldsymbol{u}_1, \boldsymbol{u}_2) = e^{-||\boldsymbol{w}_1 - \boldsymbol{w}_2||^2/2t} = e^{-r_\gamma^2 \cdot ||\tilde{\boldsymbol{w}}_1 - \tilde{\boldsymbol{w}}_2||^2/2t} = \tilde{G}_{t/r_\gamma^2}(\tilde{\boldsymbol{w}}_1, \tilde{\boldsymbol{w}}_2). \tag{3}$$

Therefore, applying Lemma 2, for any $t > 0$, we have

$$\min_{\nu \in \mathcal{M}(\mathbb{S}_r^*(\boldsymbol{v}_0))} I_{G_t}[\nu] = \min_{\nu \in \mathcal{M}(\mathbb{S}_r^*(\boldsymbol{v}_0))} \int_{\mathbb{S}_r^*(\boldsymbol{v}_0)} \int_{\mathbb{S}_r^*(\boldsymbol{v}_0)} G_t(\boldsymbol{u}_1, \boldsymbol{u}_2) \, \mathrm{d}\nu(\boldsymbol{u}_1) \, \mathrm{d}\nu(\boldsymbol{u}_2)$$

$$= \min_{\nu \in \mathcal{M}(\mathbb{S}_r^*(\boldsymbol{v}_0))} \int_{\mathbb{S}^{d-2}} \int_{\mathbb{S}^{d-2}} G_t(\phi^{-1}(\tilde{\boldsymbol{w}}_1), \phi^{-1}(\tilde{\boldsymbol{w}}_2)) \, \mathrm{d}\nu \circ \phi^{-1}(\tilde{\boldsymbol{w}}_1) \, \mathrm{d}\nu \circ \phi^{-1}(\tilde{\boldsymbol{w}}_2)$$

$$= \min_{\mu \in \mathcal{M}(\mathbb{S}^{d-2})} \int_{\mathbb{S}^{d-2}} \int_{\mathbb{S}^{d-2}} \tilde{G}_{t/r_\gamma^2}(\tilde{\boldsymbol{w}}_1, \tilde{\boldsymbol{w}}_2) \, \mathrm{d}\mu(\tilde{\boldsymbol{w}}_1) \, \mathrm{d}\mu(\tilde{\boldsymbol{w}}_2)$$

$$= \int_{\mathbb{S}^{d-2}} \int_{\mathbb{S}^{d-2}} \tilde{G}_{t/r_\gamma^2}(\tilde{\boldsymbol{w}}_1, \tilde{\boldsymbol{w}}_2) \, \mathrm{d}\sigma_{d-2}(\tilde{\boldsymbol{w}}_1) \, \mathrm{d}\sigma_{d-2}(\tilde{\boldsymbol{w}}_2)$$

$$= \int_{\mathbb{S}_r^*(\boldsymbol{v}_0)} \int_{\mathbb{S}_r^*(\boldsymbol{v}_0)} G_t(\boldsymbol{u}_1, \boldsymbol{u}_2) \, \mathrm{d}\sigma(\boldsymbol{u}_1) \, \mathrm{d}\sigma(\boldsymbol{u}_2) = I_{G_t}[\sigma],$$

indicating that the normalized surface area measure $\sigma$ uniquely minimizes the $G_t$-energy on $\mathbb{S}_r^*(\boldsymbol{v}_0)$.

Furthermore, for any $t > 0$ and each $N > 0$, denote the $N$-point minimizer of the $\tilde{G}_{t/r_\gamma^2}$-energy on $\mathbb{S}^{d-2}$ as

$$W_N^* = \{\tilde{\boldsymbol{w}}_1^*, \cdots, \tilde{\boldsymbol{w}}_N^*\} = \operatorname*{arg\,min}_{\tilde{\boldsymbol{w}}_1, \cdots, \tilde{\boldsymbol{w}}_N \in \mathbb{S}^{d-2}} \sum_{1 \le i < j \le N} \tilde{G}_{t/r_\gamma^2}(\tilde{\boldsymbol{w}}_i, \tilde{\boldsymbol{w}}_j),$$

and let $\mu_N$ denote the normalized counting measure associated with $W_N^*$. By Lemma 2, we know that $\{\mu_N\}_{N=1}^\infty$ converge weak* to $\sigma_{d-2}$. According to Equation 3, the corresponding $N$-point configuration $U_N^* = \{\phi^{-1}(\tilde{\boldsymbol{w}}_1^*), \cdots, \phi^{-1}(\tilde{\boldsymbol{w}}_N^*)\}$ minimizes the $G_t$-energy on $\mathbb{S}_r^*(\boldsymbol{v}_0)$ as

$$U_N^* = \{\phi^{-1}(\tilde{\boldsymbol{w}}_1^*), \cdots, \phi^{-1}(\tilde{\boldsymbol{w}}_N^*)\} = \operatorname*{arg\,min}_{\boldsymbol{u}_1, \cdots, \boldsymbol{u}_N \in \mathbb{S}_r^*(\boldsymbol{v}_0)} \sum_{1 \le i < j \le N} G_t(\boldsymbol{u}_i, \boldsymbol{u}_j).$$

Denoting the associated normalized counting measure as $\nu_N$, we have $\nu_N \circ \phi^{-1} = \mu_N$. Moreover, for any continuous function $f_c : \mathbb{S}_r^*(\boldsymbol{v}_0) \mapsto \mathbb{R}$, $g_c = f_c \circ \phi^{-1} : \mathbb{S}^{d-2} \mapsto \mathbb{R}$ is also a continuous function due to the continuity of $\phi^{-1}$. Therefore, we have

$$\lim_{N \to \infty} \int_{\mathbb{S}_r^*(\boldsymbol{v}_0)} f_c(\boldsymbol{u}) \, \mathrm{d}\nu_N(\boldsymbol{u}) = \lim_{N \to \infty} \int_{\mathbb{S}^{d-2}} g_c(\tilde{\boldsymbol{w}}) \, \mathrm{d}\mu_N(\tilde{\boldsymbol{w}})$$

$$= \int_{\mathbb{S}^{d-2}} g_c(\tilde{\boldsymbol{w}}) \, \mathrm{d}\sigma_{d-2}(\tilde{\boldsymbol{w}})$$

$$= \int_{\mathbb{S}_r^*(\boldsymbol{v}_0)} f_c(\boldsymbol{u}) \, \mathrm{d}\sigma(\boldsymbol{u}),$$

implying $\{\nu_N\}_{N=1}^\infty$ converge weak* to the normalized surface area measure $\sigma$ on $\mathbb{S}_r^*(\boldsymbol{v}_0)$. $\qquad\square$

## B  LINEAR SEPARABILITY OF UNIT HYPERSPHERES

Unlike Euclidean spaces, the unit hypersphere provides linear separability for well-clustered points as described in the following proposition.

**Proposition 2.** *Let $\mathbb{V}_k = \{\boldsymbol{v}_1^{(k)}, \cdots, \boldsymbol{v}_{N_k}^{(k)}\}$ denotes the set of embeddings on $\mathbb{S}^{d-1}$ corresponding to samples from the $k$-th class, and let $\mathbb{C}_k = \{\boldsymbol{v} \in \mathbb{S}^{d-1} : \boldsymbol{v}^\top \boldsymbol{c}_k \ge b_k\}$ represents some hyperspherical cap for $k \in [K]$. If embeddings from different classes are clustered on disjoint caps, that is, $\mathbb{V}_k \subset \mathbb{C}_k$ for $k \in [K]$ and $\mathbb{C}_i \cap \mathbb{C}_j = \varnothing$ for $i \ne j$, then the embeddings can be classified by a linear classifier.*

*Proof.* The proof of Proposition 1 is straightforward. Given $K$ disjoint hyperspherical caps $\mathbb{C}_k = \{\boldsymbol{v} \in \mathbb{S}^{d-1} : \boldsymbol{v}^\top \boldsymbol{c}_k \ge b_k\}$ distributed according to the class label for $k = 1, \cdots, K$, we can build a linear classifier of the form

$$f(\boldsymbol{v}) = \boldsymbol{W}\boldsymbol{v} - \boldsymbol{b},$$

where $\boldsymbol{W} = [\boldsymbol{c}_1, \cdots, \boldsymbol{c}_K]^\top$ and $\boldsymbol{b} = (b_1, \cdots, b_K)$. Then for any embedding $\boldsymbol{v} \in \bigcup_{k=1}^{K} \mathbb{V}_k$, one can make correct label classification according to the non-negative entry of $f(\boldsymbol{v})$.

$\square$

Proposition 2 suggests to enforce the embeddings to be clustered on the disjoint hyperspherical caps corresponding to their semantic labels to preserve the task-relevant information in embeddings.

## C  TRAINING DETAILS IN EXPERIMENTS

The training details regarding the GEADA framework on the Digits, CIFAR-10-C, CIFAR-100-C, PACS and Office-Home benchmarks in Section 4 are summarized in Table 6.

Table 6: Hyperparameters for training GEADA on benchmarks in Section 4.

| Hyperparameters | Digits | CIFAR-10-C | CIFAR-100-C | PACS | Office-Home |
|---|---|---|---|---|---|
| Epochs | 30 | 100 | 100 | 40 | 60 |
| Batch size | 128 | 128 | 128 | 64 | 64 |
| $\tau_{\mathrm{adv}}$ | 0.1 | 0.2 | 0.2 | 0.2 | 0.2 |
| $\lambda_{\mathrm{adv}}$ | 0.5 | 0.5 | 0.5 | 0.2 | 0.2 |
| $\tau_{\mathrm{sup}}$ | 0.3 | 0.3 | 0.3 | 0.3 | 0.3 |
| $\lambda_{\mathrm{sup}}$ | 1.0 | 1.0 | 0.5 | 1.0 | 0.5 |
| $\tau_{\mathrm{crop}}$ | - | - | - | 0.3 | 0.3 |
| $\lambda_{\mathrm{crop}}$ | - | - | - | 0.75 | 0.75 |
| Feature Extractor $f$ | | | | | |
| Optimizer | Adam | SGD | SGD | SGD | SGD |
| Learning rate | 0.001 | 0.1 | 0.1 | 0.001 | 0.001 |
| Weight decay | 0.0 | 1e-4 | 1e-4 | 5e-4 | 5e-4 |
| Linear Head $h$ | | | | | |
| Optimizer | Adam | SGD | SGD | SGD | SGD |
| Learning rate | 0.005 | 0.002 | 0.002 | 0.001 | 0.001 |
| Generative Network $G$ | | | | | |
| Optimizer | Adam | Adam | Adam | Adam | Adam |
| Learning rate | 0.01 | 0.01 | 0.01 | 0.005 | 0.005 |

