# OpenReview forum: "Generative and Explainable Data Augmentation for Single-Domain Generalization"
_ICLR.cc/2024/Conference — Submitted to ICLR 2024_

### Official Review · Reviewer_GN3R · 2023-10-31

**Soundness:** 2 fair
**Presentation:** 3 good
**Contribution:** 2 fair
**Rating:** 3
**Confidence:** 4

**Summary:**

This paper proposes Generative and Explainable Adversarial Data Augmentation (GEADA) to tackle the single-domain generalization of image classification. The core components consist of 1). an augmentor to synthesize diverse yet semantically consistent augmentations and 2). a projector to learn domain-invariant representations from the augmented samples. The augmentor comprises a generative model responsible for style augmentation, and a cropping module for explainable geometric augmentation. The theoretically-grounded contrastive called adversarial contrastive loss and supervised centroid loss are incorporated to promote the diversity of generated augmentations and the robustness of learned representations. Extensive experiments on multiple standard domain generalization benchmarks demonstrate the superiority of the proposed method.

**Strengths:**

1.	Single-domain generalization (SDG) aims to promote model generalization on unseen domains solely based on training data from a single source. The proposed GEADA algorithm shows superior performance in the SDG task compared to other methods. GEADA combines Augmentor and Projector, which are responsible for the generation of diverse but semantically consistent samples and the learning of robust representations, respectively. GEADA theoretically justified loss functions for both data augmentation and representation learning by investigating the unit hypersphere geometry.

2.	The authors integrate both style and geometric augmentation in the Augmentor. Starting from random style codes, the generative network can synthesize an arbitrary number of images that exhibit diverse styles while maintaining semantic consistency. To generate geometrically diverse views without sacrificing task-relevant information, an explainable cropping technique is proposed, which leverages a model interpretation approach.

3.	The framework incorporates two theoretically grounded loss functions, called adversarial contrastive loss and supervised centroid loss. Adversarial contrastive loss encourages a uniform distribution of different views around the source sample in the embedding space, promoting the augmented views' diversity and semantic consistency. Supervised centroid loss aligns the augmented views with the corresponding uniformly distributed class centroid on the unit hypersphere, which promotes domain-invariant representation learning.

**Weaknesses:**

1.	My main concern is the novelty and motivation of this paper. The author devises a rather cumbersome generative style modulation method, which simply changes the color of the image. Is this really necessary? I doubt that this module is really useful, and the author should compare it to typical augmentation methods such as color jitter. The module changes the color of the image in a random way without any prior, which may produce confusing results. Therefore, the author should provide visualizations of the color migration. Moreover, the experimental results in Figure 3 confirm that M=4 yields the best results, so how can the generative diversity of this module be explained?

2.	The author claims that the explainable cropping module Xcrop leverages gradients to guide the selection of cropping regions, thus maintaining diversity while ensuring that all regions are related. However, the gain from this approach may not be due to the explainable properties claimed by the authors, I think CenterCrop may have similar results, so the authors should compare it with more cropping schemes. The author should also provide corresponding visualizations to substantiate the justification of the explainable cropping module. Furthermore, the authors should perform a sensitivity analysis on the number of cropped patches P.

3.	The authors should perform more ablation experiments to justify the module, such as the color and geometric augmentation in Augmentor. Similarly, the authors should also perform an ablation of the two items of Adversarial Contrastive Loss to verify the claim for diversity and semantic consistency.

**Questions:**

Please see the weakness section above.

---

### Official Review · Reviewer_H7VV · 2023-10-31

**Soundness:** 2 fair
**Presentation:** 2 fair
**Contribution:** 2 fair
**Rating:** 5
**Confidence:** 3

**Summary:**

The paper proposed three techniques to improve the performance of single domain generalization: 1. A style modulation layer to enrich the color diversity of the source samples, 2, a gradient guided crop technique and 3, a supervised centroid loss to align the augmented views with the corresponding uniformly distributed class centroid on the unit hypersphere. The paper conducted extensive experiments to verify the proposed method.

**Strengths:**

1.	The proposed three techniques are well present and provide good theoretical deductions.
2.	The paper is well written and easy to follow.

**Weaknesses:**

1.	The proposed techniques are kind of a modification version of previous techniques, making it less novel. For example, the idea of using style modulation layer has been used in [a,b], the paper should differentiate from these methods and discuss the advantage over these methods. I understand this work is on the color of original images, making it quite limited when changing the style.
2.	Only the supervised centroid loss is studied in Fig 3b, and there is no ablation study on the first two techniques for their contributes to the final performance gain.
3.	Each technique introduces new hyper-parameters, as list in Table6, making the method hard to apply. It is valuable to provide a total loss and discuss the impact and sensitivity of these hyper-parameters.
a.	Progressive Domain Expansion Network for Single Domain Generalization, cvpr 2021
b.	Feature Stylization and Domain-aware Contrastive Learning for Domain Generalization, ACM MM 2021

**Questions:**

1.	The proposed style layer can only change the color distribution of the raw images? Then how this technique along improves the performance? Is it powerful than other style augmentation methods?
2.	Can the proposed three techniques be applied along? And it is necessary to provide an ablation study for each technique and the hyper-parameter sensitivity experiment.
3.	Only the leave-one-domain-out performance on the Office-Home benchmark is reported, how is the number on PACS? Since most works report this number, it is valuable to provide this and compare to SOTA methods.

---

### Official Review · Reviewer_ZuPr · 2023-10-31

**Soundness:** 2 fair
**Presentation:** 2 fair
**Contribution:** 2 fair
**Rating:** 5
**Confidence:** 3

**Summary:**

This paper introduces a framework, GEADA, for addressing the single-domain generalization challenge in image classification. The framework comprises two competing components, an augmentor and a projector, which are designed to work together to learn domain-invariant representations from augmented samples. The augmentor employs a generative network for style transformations and an attribution-based cropping module for geometric augmentations. The proposed loss functions aim to promote the diversity of generated augmentations and the robustness of learned representations. The authors evaluate their approach on multiple standard domain generalization benchmarks and demonstrate its effectiveness against domain shifts.

**Strengths:**

- The paper is well-structured and clearly written, making it easy to follow. Especially Figure 2 is drawn, which can help readers intuitively understand the proposed method.
- It is good to provide a theoretical basis (Proposition 1) for the loss function (Equation 1).
- The Aaugmentor designed in the paper is interesting and more promising than traditional rule-based stochastic data augmentation.

**Weaknesses:**

- The explainability of the Augmentor is not explained clearly enough for me, and I am still unclear as to why this design makes the Augmentor explainable. The paper also lacks experimental visualizations to support this claim.
- The method's significance is not strong enough. Based on the experimental results presented in this paper, the proposed method fails to outperform the baselines in several experiments.
- The work lacks necessary hyperparameter analysis experiments, such as exploring different \lambda values in the method, making it difficult to understand the principle for hyperparameter selection. Additionally, the paper lacks necessary ablation studies, such as analyzing the role of each component in the Augmentor, which would greatly benefit the study's credibility and rigor.

**Questions:**

- Why is Gaussian potential used in the loss function to measure sample similarity instead of cosine similarity or other measures? What is the guideline or reasoning behind this design?
- In Equation 1, why is it necessary to constraint that all augmented samples have the same distance from the original sample?
- It would be better to provide visualizations of learned features of the method, as it can help to intuitively understand the effect of the loss function design.
- It would be better to provide pseudocode for the algorithm, as the method involves two networks and two loss functions that seem to be interrelated. Pseudocode can help readers quickly and accurately understand the method.

---

### Meta-Review · Area_Chair_ZYR5 · 2023-12-12

**Metareview:**

This paper received originally the following negative ratings: 5, 5, and 3.
Poor organization and clarity of the paper, weak motivations and novelty aspects, insufficient experimental analysis and ablations, and poor performances wrt the baselines are among the issues raised by the reviewers.
Rebuttal was not provided by the authors.

This paper is not acceptable for publication at ICLR 2024.

**Justification For Why Not Higher Score:**

N/A

**Justification For Why Not Lower Score:**

N/A

---

### Decision · Program_Chairs · 2024-01-16

Reject